# Wettability of Wood Surface Layer Examined From Chemical Change Perspective

**Eva Annamaria Papp** [1], **Csilla Csiha** [1], **Adam Nandor Makk** [2], **Tamas Hofmann** [2] and **Levente Csoka** [3,*]

[1] Institute of Wood Based Products and Technologies, University of Sopron, Bajcsy-Zs. E. 4, 9400 Sopron, Hungary; papp.eva.annamaria@uni-sopron.hu (E.A.P.); csiha.csilla@uni-sopron.hu (C.C.)
[2] Institute of Chemistry, University of Sopron, Bajcsy-Zs. E. 4, 9400 Sopron, Hungary; makk.adam.nandor@uni-sopron.hu (A.N.M.); hofmann.tamas@uni-sopron.hu (T.H.)
[3] Institute of Cellulose and Paper Technology, celltech-paper Ltd., 9400 Sopron, Hungary
* Correspondence: csoka.levente@celltech-paper.hu

**Abstract:** The effect of artificial ageing on spruce (*Picea abies*), beech (*Fagus sylvatica* L.), birch (*Betula pendula*), and sessile oak (*Quercus petraea*) wood surfaces were investigated using qualitative (total phenolic and total soluble carbohydrate content) chemical examination methods. During ageing ($\sum$240h), the influence of surface chemistry modifications was monitored by contact angle measurements of polar, dispersive (distilled water), and dispersive (diiodomethane) liquids. The results clearly show the relation between the ratio of main chemical components of the wood surface layer and surface wettability during artificial radiation. The identified surface chemistry modifications cause more significant change in the contact angle of polar and dispersive liquid, relative to the change of dispersive liquid contact angle. Chemical changes of the wood surface layer are due to the degradation of the main wood components (cellulose, hemicelluloses, and lignin) which can be properly monitored by total phenolic (TPC) and total soluble carbohydrate content (TSCC) measurements.

**Keywords:** wettability; phenol; carbohydrates; beech; birch; spruce; sessile oak

## 1. Introduction

Regarding the antioxidant capacity of wood extracts, including phenolic compounds, numerous studies focus on the total phenolic content measurements in food research or medical biology [1]. Only a few researches deal with the investigation of extractive content of wood, while most researches report on the mechanical properties of wood. The main goal of this research was to detect the relation between changes of chemical components and the apparent contact angle of wood surface during long-term artificial radiation. To find these relations, total carbohydrate and total phenolic compound measurements were performed in parallel with contact angle measurements to identify decomposition products of the wood surface thin layer and to monitor the behavior of contact angle of the wood surface.

There are considerable differences between wood species regarding the ratio of lignin, as their structural material. While the lignin content of hardwoods is 18%–25%, for softwoods, it is between 25% and 35% generally [2]. The considerable diversity in quality and quantity of wood components is reflected in the amount of extractives. There can be significant differences in extractive content of samples coming from different species [3]. Although extractives are present only in minor quantity relative to the quantity of main chemical wood components, their influence is intense on the chemical nature of different wood species. Extractives affect the color and odor of wood, while even the efficiency of such different processing technologies as pulping and drying have and impact

on the durability, adhesion, and hygroscopic behavior of wood, as noted by Umezawa [4]. Previous researches described the relation between wood mechanical properties and phenolic compounds. Aloui et al. [5] demonstrated the effect of phenolic compounds on the durability of oak wood and defined that the higher the proportions of phenol, the higher the resistance of wood is. According to Trapp et al. [6], in terms of wood surface, the intensification of hydrophobic character can be primarily related to the increase of phenolic compounds quantity. The high degree of wood wettability can be attributed to the various hydrophilic components (e.g., hemicelluloses) of the wood surface [7]. Many studies suggest that adhesion is highly affected by extractives of wood and the wood surface layer [7–11]. High concentration extract material on wood surfaces could become a physical obstacle, making difficult the chemical bonding between, e.g., glue and wood. This type of extractive layer can decrease the surface energy and extent of wettability, and can degrade the penetration of glue into the wood material [12]. According to certain studies, there isn't a clear correlation between adhesion and the quantity of extractives [13], however Nguyen and Johns [14] stated that the percentage of wood extractive content is in inverse proportion to wettability. During contact angle measurements, wood species having the higher extractive and lignin content were found to have the higher contact angle values [15–17]. The different parts of wood due to their chemical and structural complexity can be wetted in different manner and with different success, typically decreasing with the increase of the extract materials [18]. The quantitative difference of extractives is able to cause even 40% variance with regard to wettability [19].

The wettability of wood depends on several factors due to its complexity, such as species, storage conditions (water, sunlight, biotic and abiotic factors), drying process, and cutting direction, as stated by Nguyen and Johns [20] and Kalnins et al. [21]. Wettability is also affected by the chemical composition of the surface, as stated by Kishino and Nakano [22], and the density of wood, as noticed by Amorim et al. [23].

One of the simplest methods to determine the surface energy of a solid surface is to measure the contact angle of different liquid drops on the wood surface [24]. Theoretically, the contact angle instantaneously characterizes the feature of the solid-liquid system and is measurable in a given condition [25]. Assuming that instantaneous contact angle changes are consequence of hydrodynamic processes, it could be stated hypothetically that its most characteristic value can be measured in that moment, when the shape of the liquid droplet is formed on the surface (or at a very close time) [26]. Behavior of a liquid drop relaxing on a solid surface are influenced by the changes of the wood surface [27]. The reason for this is that the contact angle is a phenomenological parameter which is not influenced just by surface energy, but by surface roughness, surface heterogeneity, moisture content, and a lot of other factors of the wood material [28].

The photodegradation of wood makes it more complicated to find the influencing factors of wettability. Those research results are relevant for us which deal with the chemical changes of wood surface caused by artificial ageing, and with the influence of radiation-caused degradation and its effect on contact angle.

From the adhesion point of view, one of the most critical factors is the time elapsed since machining. In the case of wood, the most optimal surfaces for gluing and surface treatments are freshly prepared surfaces [9,29]. Accordingly, the wettability of different wood species decreases in parallel with the age of machined surfaces because of the chemical transformation of extractives on wood surfaces [30]. Wood surfaces show significant changes in their surface free energy even after days of conditioning, as reported in different studies [31,32]. Lignin is easily oxidisable by photodegradation and its structural changes can be detected right when it comes in contact with the air, or during long-term storage [33]. Surface changes of wood due to natural aging can be imitated using artificial ageing apparatus under laboratory conditions. Both processes can be blamed for decreasing the surface energy parameter, caused by the migration of hydrophobic extractives onto the surface of wood [34].

It is noted that lignin together with extractives is able to increase the hydrophobic character of wood surface. Unlike the cellulose and hemicellulose, lignin has relatively hydrophobic character [35].

Lignin is able to play a role in the increase of hydrophobicity, being active in moisture transport in wood [36,37]. Williams et al. [38] stated that light irradiation causes rapid colour changes followed by processes having a strong impact on surface wettability in the case of some wood species. Similarly, to other natural polymers lignin and polyphenols, as wood components, absorb UV radiation. Due to UV absorption, photolytic, photo-oxidative, and thermo-oxidative reactions start to develop in wood material and those reactions are responsible for the degradation caused by sunlight [39]. Radicals formed due to the photodegradation of lignin protect lignin and also the complex wood system from further photodegradation effects [40]. Besides lignin, the most intensive visible changes due to photodegradation are caused by extractives. The reason of this is that extractives have strong light absorption attribute [41,42] and in this manner protect the main wood components [43]. During artificial or natural ageing hydrophobic extractives migrate to the wood surface and cause decrease of the wettability [34]. The dissolution of phenolic compounds is result of surface structure also, which is typical when the wood surface comes into contact with liquids [44–47]. In case of some wood species due to artificial ageing wettability increases. The phenomenon of ablation caused by UV radiation is the reason for the mitigation of materials able to decrease wettability of wood surface [48]. Due to long-term UV radiation of the cell wall, researchers [49] detected significant quantity of water-soluble decomposition products, which can alter the quality of wettability.

Penetration depth of UV radiation and visible light are different, and the photodegradation caused by them is of different measure also. Based on Williams [50] and Pandey [41] the usual penetration depth of UV radiation is ~75 μm. These values depend on the density of wood and ratio of earlywood and latewood. Hon and Ifju [51] stated that the maximum UV penetration depth in wood is not more than 80 μm. According to the research of Németh and Faix [52], the upper 75-μm thin layer of wood should be monitored when examining photodegraded wood surfaces.

The present investigations are directed towards the development of a novel method for the evaluation of wood surface chemical compound changes under artificial ageing in the case of different wood species.

## 2. Materials and Methods

During the investigation of artificially aged wood surfaces, contact angle measurements (using distilled water and diiodomethane), moisture content (MC) examinations, total phenolic and total soluble carbohydrate content measurements were performed on wood samples of beech (*Fagus sylvatica* L.), birch (*Betula pendula*), sessile oak (*Quercus petraea*), and spruce (*Picea abies*), as depicted in Figure 1.

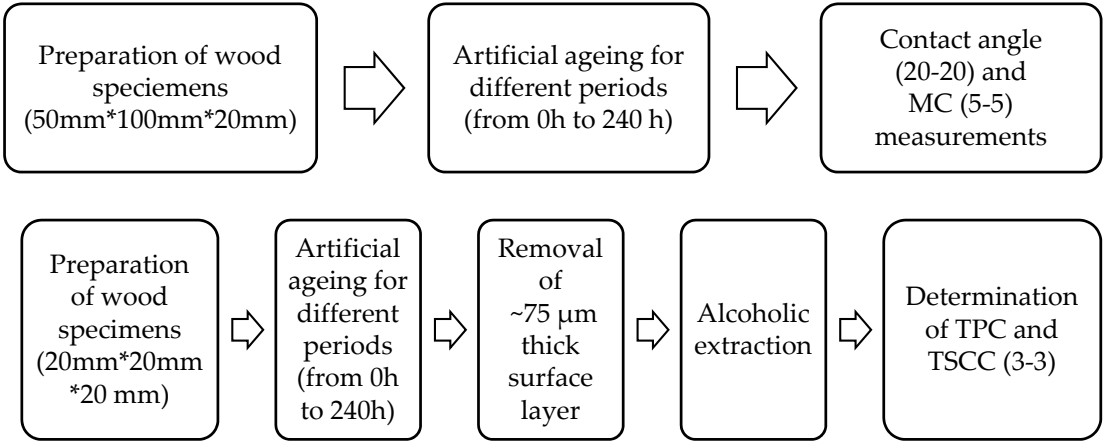

**Figure 1.** Flow diagram of experimental procedures.

*2.1. Preparation of Wood Specimens*

Before ageing, all wood surfaces were sanded with sandpaper of grit size 150. From the different wood species similar samples were prepared for different measurements, including the specimens

for contact angle and moisture content examinations from different boards with dimensions $50 \times 100 \times 20$ mm$^3$ and specimens for total phenol and total carbohydrate content measurements with dimensions $20 \times 20 \times 20$ mm$^3$.

### 2.2. Technological Background of Artificial Ageing

The artificial ageing of different wood species was performed with Original Hanau Suntest (HERAEUS, Hanau, Germany) equipped with xenon bulb and built-in UV mirror, started within 1 h after machining. The different measurements were performed in the following ageing periods: 0 h (control), 1 h, 3 h, 5 h, 8 h, 10 h, 15 h, 20 h, 30 h, 60 h, 96 h, 132 h, 174 h, and 240 h, respectively. During artificial ageing, the temperature of wood surfaces was monitored by using Maxwell MX 25-903 type (GLOBIZ, Ashford, Kent, UK) digital thermometer. Temperature monitoring was responsible for the chamber temperature do not exceed the $55 \pm 5$ °C value which was defined as maximum reachable ageing temperature. Based on practical experience of artificial ageing the temperature over 60 °C can induce major surface chemical changes.

For moisture content measurements HM8-WS5 Merlin Moisture Meter (MERLIN Technology GmbH, Tumeltsham, Austria) was used, with ~5 mm scanning depth, and ~20 cm$^2$ maximum measuring area to perform 5 moisture content examinations on each wood surface after ending different ageing periods.

### 2.3. Contact Angle Measurements of Artificially Aged Wood Surfaces

Dynamic contact angle measurements were performed on each wood surface by using PG-X goniometer (FIBRO SYSTEMS AG, Hagersten, Sweden): 20 by using distilled water and 20 by using diiodomethane (SIGMA-ALDRICH, St. Louis, MO, USA). Measurements were performed in case of all ageing periods, 1 sec after the drop release and formation of the liquid droplet on the surface, as previously agreed [26].

### 2.4. Alcoholic Extraction and Determination of Total Phenolic and Total Soluble Carbohydrate Contents

To measure the total phenolic and total soluble carbohydrate contents, a ~75-μm thick layer was collected by alcohol sterilized steel blade from sample surfaces, following previously agreed ageing periods.

For alcoholic extraction methanol: water (volume ratio 4:1) (with methanol from REANAL, Budapest, Hungary) mixture was used and BRANSON 3510 ultrasonic bath (EMERSON, St. Louis, MO, USA) was applied for 20 min, after 5 mL extracting agent was mixed with the previously collected wood material (~0.05 g). The extract produced in this manner was centrifuged with a MiniSpin spinner (EPPENDORF, Hamburg, Germany) for 10 min (13,400 rpm).

The total phenolic content (TPC) was calculated based on the Folin–Ciocalteu method [53]. First, Folin–Ciocalteu reagent (VWR International, Debrecen, Hungary) (2.5 mL, tenfold dilution) was mixed with the extract (0.5 mL), and after 1 min application time, 2 mL (with concentration 0.7 M) $Na_2CO_3$ (VWR International, Debrecen, Hungary) solution was blended with the mixture. The reaction mixture was warmed in Memmert WNB 200 water bath (MEMMERT GmbH, Buechenbach, Germany) at 50 °C for 5 min. After warming in water bath, the solutions were cooled in cold water bath, until the temperature of solutions reached ~25 °C. For determination of total phenol content Metertech SP 8001 spectrophotometer (METERTECH Inc., Taipei, China) was used at 760 nm wavelength and as standard, quercetin (SIGMA-ALDRICH, St. Louis, MO, USA) was chosen.

Total soluble carbohydrate content (TSCC) was calculated based on method of Dubois et al. [54]. Firstly, phenol solution (REANAL, Budapest, Hungary) (0.5 mL; dilution: 5%) was mixed to the extract (0.5 mL). After mixing to the solution 2.5 mL concentrated sulfuric acid (REANAL, Budapest, Hungary), sealed test tubes were hold for 10 min at room temperature, followed by a second cooling for 20 min, in a 25 °C temperature water bath. Total soluble carbohydrate content was measured by using Metertech SP 8001 spectrophotometer (METERTECH Inc., Taipei, China) at 490 nm. During the total soluble

carbohydrate content measurements, glucose was used as standard (SIGMA-ALDRICH, St. Louis, MO, USA). For the determination of TPC and TSCC, 3-3 replicates were analyzed.

*2.5. Statistical Analysis*

Statistical analysis was performed in Table 1, namely the analysis of variance (ANOVA) and t-test in order to explore significant variations between the following groups (measured data). No real significant differences between groups were considered in the ANOVA test when the value of statistic was close to l and relative low statistic value in case of t-test.

**Table 1.** The statistical analysis on each measured parameter.

| Beech, Spruce, Birch, Sessile oak | †DWCA-TFC |
| | DWCA-TSCC |
| | ‡DMCA-TFC |
| | DMCA-TSCC |
| | TFC-TSCC |
| | DWCA-DMCA |

†DWCA: Distilled water contact angle, ‡DMCA: Diiodomethane contact angle.

Datasets are given in publicly available database.

## 3. Results

*3.1. Contact Angle Changes due to Artificial Ageing*

The changes occurring in the development of the contact angles measured with different test liquids are different (bi-distilled water and diiodomethane) during ageing. The character of the development of contact angles is similar on the different wood species (Figure 2).

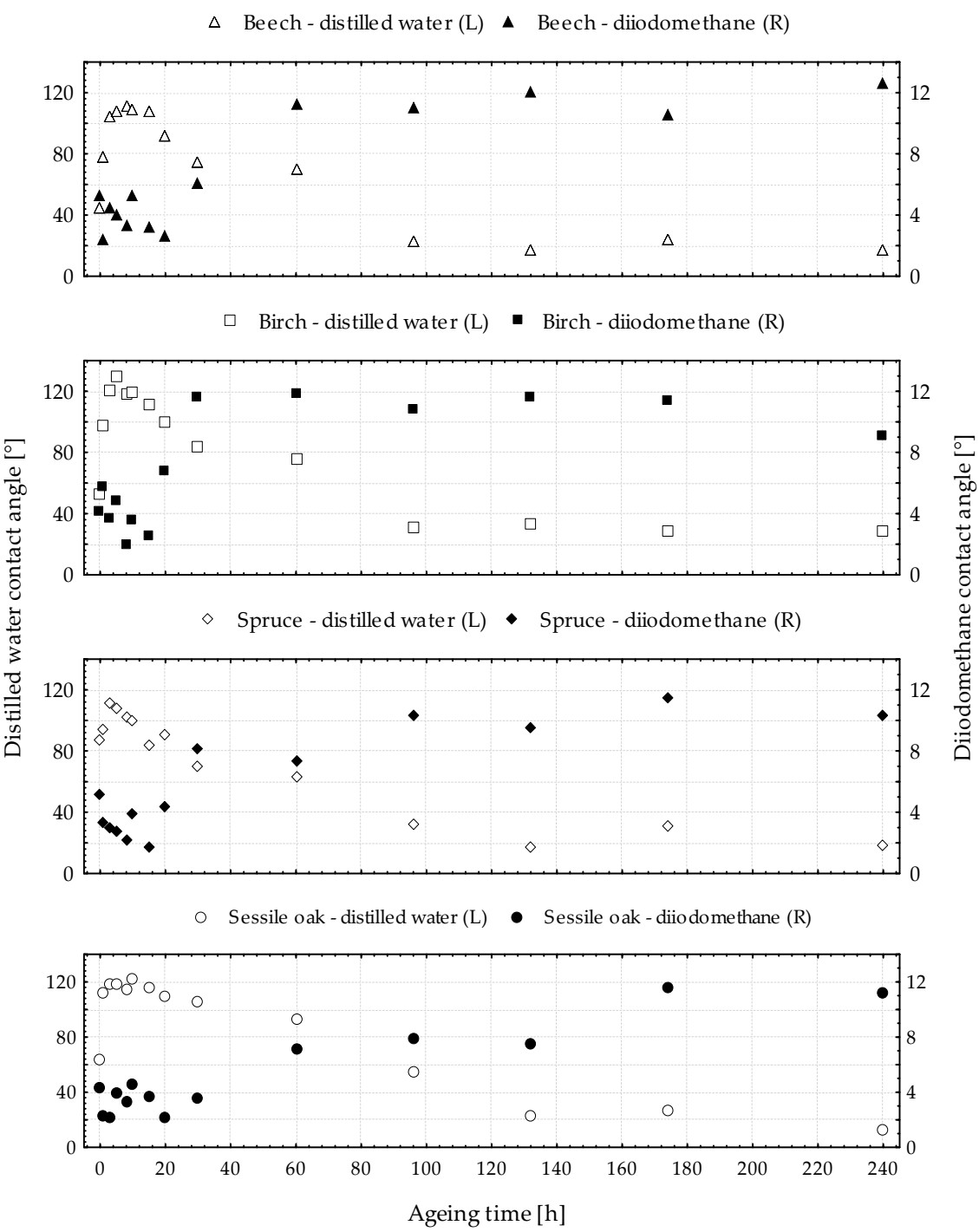

**Figure 2.** **Figure 2**. Contact angle values of distilled water and diiodomethane during artificial ageing, L—left y-axis, R—right y-axis.

　　　There is some difference within the control values (0h) of distilled water contact angles measured on different wood species (Figure 2 and Table 2). Higher contact angle values were detected on sessile oak material, associated with higher extractive content, according to previous studies [15–17]. Since it has significant influence on liquid contact angle results, moisture content determinations were performed in order to monitor wood surface layer MC changes due to artificial ageing. MC results clearly show that, after the first three hours of artificial radiation, there was no difference between two consecutive measurements, in the case of all wood species. The results clearly show the process of changes during the total duration of artificial ageing, which takes place similarly in the case of all

wood species: contact angle of distilled water increases in the first 8 h. Consequently, until the end of ageing (240 h), it has a slightly decreasing tendency.

**Table 2.** The average values of contact angle and its spread.

| Ageing Time [h] | Distilled Water Contact Angle Values [°] | | | | | | | |
| | Beech | | Spruce | | Birch | | Sessile oak | |
| | $\bar{x}$ | SD | $\bar{x}$ | SD | $\bar{x}$ | SD | $\bar{x}$ | SD |
| 0 | 45.30 | 3.86 | 87.05 | 5.33 | 52.91 | 3.19 | 63.30 | 3.82 |
| 1 | 78.61 | 3.12 | 93.69 | 5.07 | 97.84 | 9.24 | 112.47 | 2.52 |
| 3 | 104.95 | 3.42 | 111.83 | 5.54 | 120.22 | 4.01 | 118.81 | 3.43 |
| 5 | 107.97 | 2.68 | 107.83 | 3.10 | 129.43 | 3.91 | 118.57 | 1.61 |
| 8 | 111.92 | 3.08 | 101.96 | 2.78 | 118.83 | 4.32 | 114.48 | 3.18 |
| 10 | 109.05 | 3.21 | 100.39 | 3.08 | 119.92 | 4.31 | 122.64 | 2.30 |
| 15 | 108.02 | 2.59 | 83.55 | 6.45 | 111.37 | 10.55 | 115.53 | 3.05 |
| 20 | 91.98 | 3.55 | 90.23 | 4.15 | 100.21 | 5.47 | 109.54 | 3.43 |
| 30 | 74.15 | 5.21 | 70.41 | 5.27 | 84.34 | 2.34 | 105.79 | 2.93 |
| 60 | 70.11 | 3.19 | 63.00 | 2.63 | 76.09 | 2.91 | 93.49 | 7.55 |
| 96 | 23.58 | 4.29 | 32.33 | 5.63 | 31.00 | 4.98 | 54.97 | 8.66 |
| 132 | 17.72 | 1.39 | 17.42 | 1.68 | 32.87 | 3.32 | 23.15 | 2.72 |
| 174 | 24.46 | 3.94 | 31.34 | 3.92 | 28.79 | 3.95 | 26.48 | 3.09 |
| 240 | 17.09 | 1.68 | 19.03 | 2.20 | 28.48 | 3.58 | 13.44 | 2.79 |
| Ageing time [h] | Diiodomethane Contact Angle Values [°] | | | | | | | |
| | Beech | | Spruce | | Birch | | Sessile oak | |
| | $\bar{x}$ | SD | $\bar{x}$ | SD | $\bar{x}$ | SD | $\bar{x}$ | SD |
| 0 | 5.24 | 1.11 | 5.23 | 0.80 | 4.19 | 0.44 | 4.34 | 0.75 |
| 1 | 2.36 | 0.54 | 3.30 | 0.69 | 5.80 | 1.37 | 2.35 | 0.68 |
| 3 | 4.54 | 1.03 | 2.97 | 0.95 | 3.68 | 1.03 | 2.13 | 1.00 |
| 5 | 3.98 | 1.14 | 2.81 | 0.63 | 4.79 | 0.87 | 4.00 | 0.63 |
| 8 | 3.39 | 0.68 | 2.19 | 0.63 | 1.97 | 0.71 | 3.39 | 0.63 |
| 10 | 5.28 | 0.71 | 3.94 | 0.84 | 3.51 | 0.70 | 4.57 | 0.65 |
| 15 | 3.24 | 0.76 | 1.79 | 0.75 | 2.48 | 0.70 | 3.76 | 0.74 |
| 20 | 2.70 | 0.69 | 4.34 | 1.66 | 6.75 | 1.33 | 2.11 | 0.56 |
| 30 | 6.12 | 1.17 | 8.19 | 0.71 | 11.62 | 1.33 | 3.59 | 1.36 |
| 60 | 11.28 | 1.51 | 7.38 | 1.25 | 11.85 | 1.58 | 7.15 | 0.52 |
| 96 | 10.99 | 1.51 | 10.31 | 1.32 | 10.77 | 1.11 | 7.96 | 0.58 |
| 132 | 12.01 | 1.33 | 9.59 | 1.03 | 11.63 | 1.11 | 7.50 | 0.61 |
| 174 | 10.52 | 1.99 | 11.51 | 1.72 | 11.33 | 1.86 | 11.63 | 0.97 |
| 240 | 12.60 | 1.54 | 10.33 | 1.54 | 9.04 | 1.58 | 11.22 | 1.62 |

According to control values (0 h) and finally measured (after 240 h artificial radiation) distilled water contact angle values of different wood species can be concluded that the control values (0 h) are higher in case of all four wood species. Those results lead to the conclusion that higher wettability is characteristic to the wood surfaces at the end of artificial ageing (Figure 2). In addition, evaluating the total duration of artificial ageing, contact angle of both polar and disperse distilled water changes to a greater extent than the contact angle of the solely disperse diiodomethane.

### 3.2. Total Phenolic and Total Soluble Carbohydrate Content of Artificially Aged Wood Surfaces

Total phenolic content examinations were performed to detect quantity changes of phenolic extractive compounds of different wood materials (Figure 3). Phenolic compounds significantly influence the wetting of polar and disperse liquid drops, together with the wettability of wood surfaces.

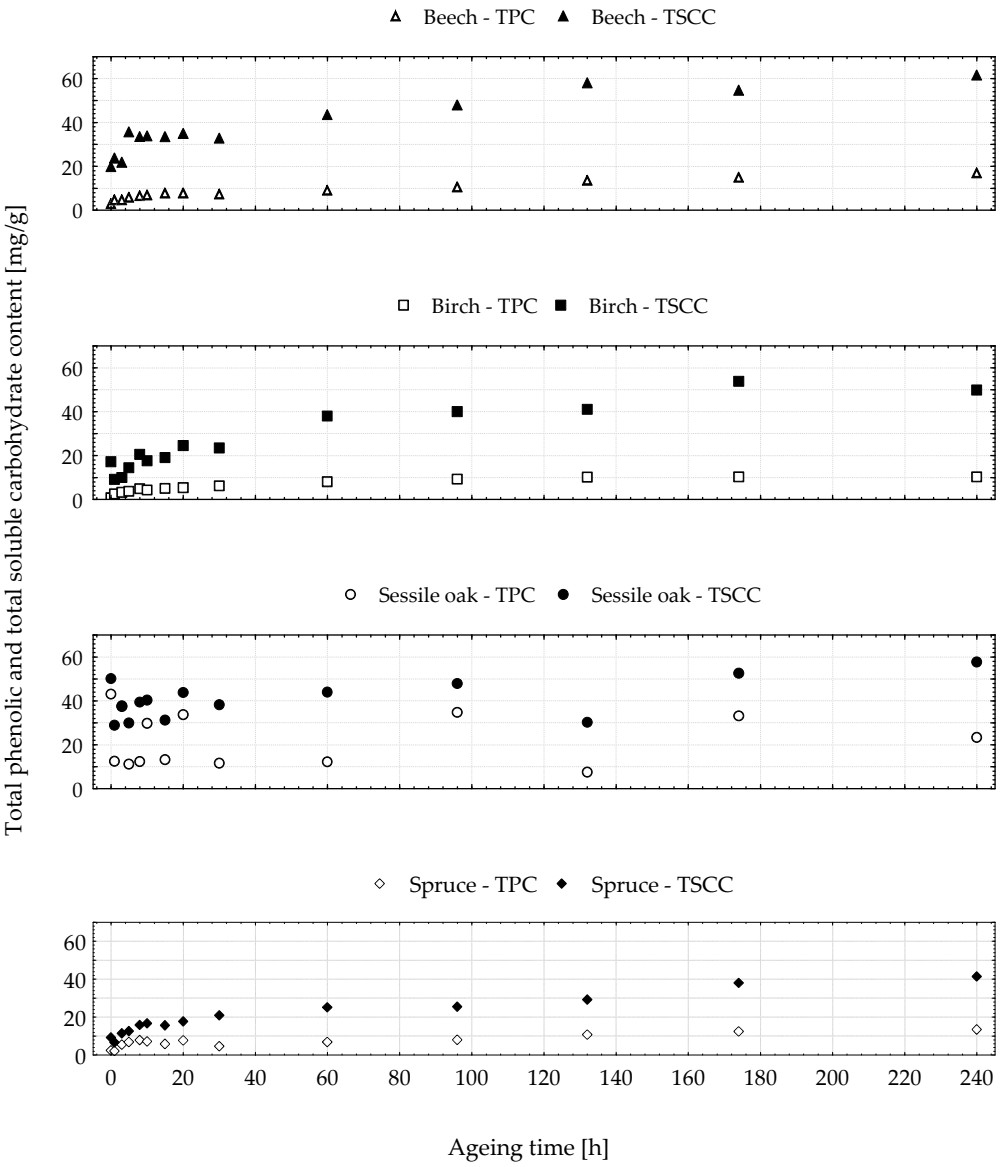

**Figure 3.** Total phenolic and total soluble carbohydrate contents as function of artificial ageing.

The results of total phenolic and total soluble carbohydrate content examinations show major difference in their measure. Total phenolic content of different wood species, except sessile oak, increases from a value about 1 mg/g to about 15 mg/g, regarding the total 240 h of artificial radiation. Parallel with the changes of total phenolic content, the total soluble carbohydrate content (for all four wood species) increases from a value of 7 mg/g to about 45–60 mg/g. The results clearly show that the carbohydrate content changes in 3–4 times higher measure than the phenolic compounds (Table 3).

**Table 3.** The average values of TPC and TSCC and its spread.

| Ageing Time [h] | TFC Values [mg/g] | | | | | | | |
| | Beech | | Spruce | | Birch | | Sessile oak | |
| | $\bar{x}$ | SD | $\bar{x}$ | SD | $\bar{x}$ | SD | $\bar{x}$ | SD |
| 0 | 2.923 | 0.042 | 2.519 | 0.096 | 0.729 | 0.083 | 43.070 | 2.842 |
| 1 | 4.706 | 0.085 | 2.127 | 0.092 | 2.533 | 0.101 | 12.481 | 1.517 |
| 3 | 4.691 | 0.042 | 5.479 | 0.155 | 3.325 | 0.095 | 37.380 | 1.187 |
| 5 | 5.974 | 0.178 | 6.805 | 0.257 | 3.665 | 0.066 | 11.201 | 0.090 |
| 8 | 6.702 | 0.055 | 7.780 | 0.205 | 4.910 | 0.183 | 12.311 | 0.466 |
| 10 | 6.867 | 0.191 | 7.119 | 0.227 | 4.328 | 0.117 | 29.842 | 1.669 |
| 15 | 7.770 | 0.072 | 5.775 | 0.253 | 5.017 | 0.136 | 13.215 | 1.001 |
| 20 | 7.819 | 0.113 | 7.732 | 0.175 | 5.328 | 0.199 | 33.759 | 1.891 |
| 30 | 7.295 | 0.139 | 4.683 | 1.907 | 6.268 | 0.330 | 11.653 | 1.045 |
| 60 | 9.113 | 0.123 | 6.807 | 0.439 | 8.108 | 0.258 | 12.208 | 0.413 |
| 96 | 10.615 | 0.136 | 8.003 | 0.478 | 9.309 | 0.261 | 34.748 | 0.870 |
| 132 | 13.613 | 0.196 | 10.804 | 0.244 | 10.095 | 0.114 | 7.503 | 0.239 |
| 174 | 14.963 | 0.370 | 12.418 | 0.886 | 10.208 | 0.254 | 33.061 | 0.905 |
| 240 | 17.041 | 0.199 | 13.371 | 0.692 | 10.281 | 0.401 | 23.380 | 0.495 |
| Ageing Time [h] | TSCC Values [mg/g] | | | | | | | |
| | Beech | | Spruce | | Birch | | Sessile oak | |
| | $\bar{x}$ | SD | $\bar{x}$ | SD | $\bar{x}$ | SD | $\bar{x}$ | SD |
| 0 | 19.762 | 0.504 | 9.255 | 0.402 | 17.177 | 2.157 | 50.172 | 3.595 |
| 1 | 23.731 | 4.575 | 6.773 | 0.755 | 9.078 | 0.829 | 28.889 | 0.976 |
| 3 | 21.906 | 2.880 | 11.468 | 1.086 | 10.058 | 2.493 | 37.681 | 7.343 |
| 5 | 35.616 | 5.721 | 12.569 | 2.080 | 14.373 | 2.557 | 29.903 | 3.324 |
| 8 | 33.638 | 2.732 | 15.849 | 2.621 | 20.624 | 1.430 | 39.438 | 10.706 |
| 10 | 33.858 | 2.186 | 16.681 | 2.880 | 17.628 | 0.174 | 40.276 | 0.554 |
| 15 | 33.459 | 3.573 | 15.555 | 3.330 | 19.073 | 0.561 | 31.259 | 2.626 |
| 20 | 34.883 | 0.163 | 17.769 | 3.698 | 24.459 | 0.950 | 43.858 | 2.782 |
| 30 | 32.876 | 5.870 | 20.903 | 3.423 | 23.505 | 3.657 | 38.251 | 5.158 |
| 60 | 43.517 | 3.090 | 25.110 | 2.013 | 37.897 | 2.891 | 44.040 | 0.488 |
| 96 | 47.914 | 1.348 | 25.370 | 7.367 | 39.954 | 0.797 | 47.991 | 1.972 |
| 132 | 58.020 | 3.653 | 29.226 | 4.664 | 41.100 | 3.795 | 30.234 | 1.348 |
| 174 | 54.711 | 1.643 | 38.010 | 1.481 | 53.740 | 5.197 | 52.610 | 6.920 |
| 240 | 61.471 | 6.214 | 41.353 | 8.009 | 49.826 | 2.215 | 57.729 | 3.781 |

## 4. Discussion

The quantitative change described above is consistent regarding the change of the measured contact angle values. The measured contact angle, TPC and TSCC values in case of different wood species show similar trend under the entire artificial ageing period.

Despite the fact that the total phenol content increases almost 15-times during 240 h of artificial xenon radiation and the total soluble carbohydrate content increases only 6–8 times, the substantially higher presence of carbohydrate has a greater influence on the contact angle of the liquid drops. Based on the decrease of contact angles an improved wood surface wettability can be concluded.

The most reactive wood components during photodegradation are the lignin and the extractives [39,41,42]. The reason for contact angle changes in the first 15–20 h of artificial radiation is that the decomposition of lignin and extractives begins, and the total phenolic content increases. In the subsequent radiation interval begins the decay of cellulose and hemicelluloses, which are less sensitive to photodegradation. The total soluble carbohydrate content increases compensate the effect of hydrophobic phenolic compounds and this is manifest in the diminution of strong variation in the contact angle values. As cellulose and hemicelluloses are present on the wood surface (and in the whole wood material) in essentially larger quantities, their hydrophilic nature is strongly manifested

in the next radiation interval proved, also by decreasing contact angle values. The observed chemical changes were further investigated with Fourier-transform infrared spectroscopy and these results will be published in the near future.

Overall, the performed t-test statistical analysis resulted in values close to 0 in each combination, which indicate that similarity exists between the measured data sets, and supports the observation that the contact angle value, for example, is susceptible to follow surface chemical changes. The ANOVA test also indicates that there is a real difference between the tested, measured data groups in each combination, as the value was close to zero.

## 5. Conclusions

Chemical changes of the wood surface layer, which can be well monitored by using total phenolic and soluble carbohydrate content examinations, are influenced by the photodegradation of the main wood components: cellulose, hemicelluloses, and lignin. According to test results of TPC and TSCC experiments, by using the 75-μm thin layer of wood specimens, both phenolic and carbohydrate content alter to a great extent due to photodegradation. The quantity increases of total soluble carbohydrates, which gives information about the degradation of cellulose and hemicelluloses, and their basically greater presence in wood material, influences the contact angle of liquid drops more than the total phenolic compounds.

To our best knowledge, there is no article that reports on TPC and TSCC evaluation considering the surface detached wood particles observed under artificial ageing. The present study shows that the preparation method and evaluation of TPC and TSCC can be significant. This could be a great advantage and novel technique, thus representing a good contribution to the surface science of wood. In that sense, our attempt to draw attention to the method used might also be useful to researchers from the other fields of the wood science, like surface treatment and the gluing of wood.

**Author Contributions:** Conceptualization, E.A.P. and L.C.; methodology, T.H. and L.C.; software, E.A.P. and A.N.M; validation, E.A.P., A.N.M. and T.H.; formal analysis, C.C.; investigation, E.A.P. and A.N.M.; resources, C.C.; data curation, L.C.; writing—original draft preparation, E.A.P.; writing—review and editing, E.A.P. and L.C.; visualization, E.A.P. and A.N.M.; supervision, C.C.; project administration, C.C.; funding acquisition, L.C. All authors have read and agreed to the published version of the manuscript.

**Funding:** This research was supported by the European Union and the State of Hungary, co-financed by the European Social Fund in the framework of TAMOP 4.2.4. A/2-11-1-2012-0001 National Excellence Program.

**Acknowledgments:** This article was made in frame of "the EFOP-3.6.1-16-2016-00018—Improving the role of research + development + innovation in the higher education through institutional developments assisting intelligent specialization in Sopron and Szombathely". IKEA Industry Sopron Ltd. for wood material supply and the carpenter workshop team of US-SKF for their skillful work are gratefully acknowledged.

**Conflicts of Interest:** The authors declare no conflict of interest.

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
