# Peer review of "Wettability of Wood Surface Layer Examined From Chemical Change Perspective"

_coatings, doi:10.3390/coatings10030257_

Round 1

Reviewer 1 Report

please see my comments in the attached file.

Author Response

Response to reviewers (Ms. Ref. No.: Coatings-735878)

The manuscript was revised according to 1. reviewer’s suggestions. In the text of the revised manuscript the changes made are marked in red.

Reviewer#1:

Comment: In Line 38, put the reference number in the end of the sentence and re-write this.

Answer: It has been corrected.

Comment: In Line 43, use affect instead of effect.

Answer: It has been corrected.

Comment: In Line 116: Please, in the end of the introduction session, add a paragraph to describe the scope of the present study and to emphasise on it’s the innovative part.

Answer: It has been added and clarified.

Comment: In Line 118: Please, add a flow diagram as figure 1, to graphically explain-describe the experimental procedure.

Answer: It has been added.

Comment: In Line 122: Please number the sub-sessions and mention the number of replicates applied. You used an extremely large number of sub-sessions. Try to minimize these to a reasonable number.

Answer: It has been corrected.

Comment: In Line 131: Provide some photos of artificial ageing.

Answer: We did not take any photo about the artificial ageing.

Comment: In Line 143: Specify the number of replicates.

Answer: It has been added at each measurement as well as can be found on Fig. 1.

Comment: In Line 200: Please omit these three lines.

Answer: Those has been omitted.

Comment: In Line 208: This is indeed a very complicated figure!! You did not even mention the number of replicates and you did not show a standard deviation or a standard error.

Answer: The number of replicates has been included and clarified in the Materials and Methods section. Standard deviation and average values can be found in the tables. Moreover, new figures has been added to clarify the complicated figure.        

Comment: In Line 209: I did not see a comment on statistical analysis. Have you identified significant differences?

Answer: Statistical analysis did not perform, but the average values and standard deviation are reported in the new tables.

Comment: In Line 218: Where this come from?

Answer: These can be seen on Fig. 2 and 3., and 63.7% of total surface tension of water is polar, which have this effect, but further clarification needs to prove that.

 Comment: In Line 261: Same comment as fig. 1.

Answer: New figure has been added to clarify that complicated figure as well.

Comment: In Line 265: Where this come from? Do you perform a statistical analysis?

Answer: It has been clarified.

Comment: In Line 271: This is an extremely poor discussion. You did not mention a single reference!!! I think it has totally to be rewritten.

Answer: Further references and clarification have been added.

Comment: In Line 288: Omit this!!

Answer: This sentence has been cancelled.

Comment: In Line 293: Not sounds conclusions.

Answer: More clarification has been added. We thank the reviewer for the comments on the manuscript.

Reviewer 2 Report

The manuscript presents the results of an investigation on the effect of artificial ageing on a wood surface (of four different wood species) using qualitative chemical methods including total phenolic and soluble carbohydrate content examinations, and describes how the resulting changes in wood chemical composition affect contact angle.

I have some questions and comments on the text:

  • In line 37, the Authors mention “ratio”, but in the next sentence they describe the lignin content – perhaps it would be better to be more precise and focus consequently on the ratio of the wood polymers or their content in the paragraph. Why is only the lignin content given here, if in the preceding sentence all wood polymers are mentioned? Please, add the values for other wood polymers or explain why is lignin so crucial in this case.
  • Line 49-50 – is the increase of wood hydrophobicity an effect of an increase or decrease in the content of phenolic compounds?
  • Line 56-57 – do the Authors mean cohesion or adhesion in the context of wettability of wood surface?
  • Line 93 – The statement that lignin has hydrophobic character is not entirely correct. It is much more hydrophobic than other main wood components (cellulose and hemicelluloses), but it also contains some hydrophilic groups. We can say that it is relatively hydrophobic in comparison with polysaccharides. Besides, in the next sentence, the Authors say that lignin is active in moisture transport in wood, which confirms that it must contain some hydrophilic groups.
  • How many replicates of each type were used for contact angle measurements? What was the standard deviation of the results obtained for each type of wood?
  • What were the conditions of artificial ageing?
  • Figure 2 – there is no “empty triangles” – the results for beech-TPC in the picture (they are marked as an extra set of circles, I guess).
  • It would be good to show the correlation between the content of phenolic/carbohydrate compounds content and contact angle values in a graph and discuss how the initial chemical composition of particular wood species influenced the changes in wettability of wood surface after artificial ageing.
  • Lines 278-286 – could the Authors refer to any research/papers to confirm the statements made in this paragraph, please?
  • Could the Authors draw any practical conclusions from the results obtain? Is the method applied for monitoring changes in the wood surface a novelty? Could be the results obtained somehow useful for modelling the chemical changes in wood upon photodegradation?

There is a lot of minor linguistic mistakes (including punctuation, “a/an” and “the” articles missing etc.) that should be corrected, e.g.:

Line 31-32 – „..report on and mechanical properties…” – remove „and”,

Line 33 – “components” instead of “composites”,

Line 40 – “extractives” instead of “extract materials”,

Line 42 – “affect” instead of “effect”,

Line 43-44 – “the color of wood” instead of “the color of wood species” etc.

Author Response

Reviewer#2:

The manuscript was revised according to the 2. reviewer’s suggestions. In the text of the revised manuscript the changes made are marked in green.

Comment: The manuscript presents the results of an investigation on the effect of artificial ageing on a wood surface (of four different wood species) using qualitative chemical methods including total phenolic and soluble carbohydrate content examinations, and describes how the resulting changes in wood chemical composition affect contact angle.

Answer: We thank the reviewer for the encouraging comments on the manuscript.

Comment: In line 37, the Authors mention “ratio”, but in the next sentence they describe the lignin content – perhaps it would be better to be more precise and focus consequently on the ratio of the wood polymers or their content in the paragraph. Why is only the lignin content given here, if in the preceding sentence all wood polymers are mentioned? Please, add the values for other wood polymers or explain why is lignin so crucial in this case.

Answer: That sentence has been corrected. The importance of lignin content has been emphasized in several places of the text. We thank to the reviewer to draw our attention to that point.   

 Comment: Line 49-50 – is the increase of wood hydrophobicity an effect of an increase or decrease in the content of phenolic compounds?

Answer: It has been corrected: “the intensifying of hydrophobic character can be primarily related to the increase of phenolic compounds quantity.   

Comment: Line 56-57 – do the Authors mean cohesion or adhesion in the context of wettability of wood surface?

Answer: We thank to the reviewer to draw our attention to that point as well, it has been corrected and changed to “adhesion”.

Comment: Line 93 – The statement that lignin has hydrophobic character is not entirely correct. It is much more hydrophobic than other main wood components (cellulose and hemicelluloses), but it also contains some hydrophilic groups. We can say that it is relatively hydrophobic in comparison with polysaccharides. Besides, in the next sentence, the Authors say that lignin is active in moisture transport in wood, which confirms that it must contain some hydrophilic groups.

Answer: We thank to the reviewer for this comment, that sentence has been changed accordingly.  

Comment: How many replicates of each type were used for contact angle measurements? What was the standard deviation of the results obtained for each type of wood?

Answer: The number of replicates has been included and clarified in the Materials and Methods section. Standard deviation and average values can be found in the tables.        

Comment: What were the conditions of artificial ageing?

Answer: More clarification of artificial ageing has been included in the Materials and Methods section.

Comment: Figure 2 – there is no “empty triangles” – the results for beech-TPC in the picture (they are marked as an extra set of circles, I guess).

Answer: Values of beech TPC are marked in Figure 3. with empty triangles. It can be found under 20 mg/g during 240 hours of ageing.

Comment: It would be good to show the correlation between the content of phenolic/carbohydrate compounds content and contact angle values in a graph and discuss how the initial chemical composition of particular wood species influenced the changes in wettability of wood surface after artificial ageing.

Answer: Tables has been added to the manuscript containing the phenolic/carbohydrate compounds content and contact angle values for the wood species. Several type of correlation trends has been fitted to the values and resulted poor R2 (0.3-0.8). The trends also did not resulted uniform fitted curve, so it seems more results should be generated to establish clear correlation, if there is any. Fitted curve for one wood species resulted second order polynomial trend and the other case 6th order or logarithmic curve showed better R2. It could be quite difficult to interpret such results but at the same time it does not exclude the possibility that there is no correlation. Statement about this has been included in the manuscript.  

Comments: Lines 278-286 – could the Authors refer to any research/papers to confirm the statements made in this paragraph, please?

Answer: References has been added to that part, but most of these statements concluded from the results.

Comment: Could the Authors draw any practical conclusions from the results obtain? Is the method applied for monitoring changes in the wood surface a novelty? Could be the results obtained somehow useful for modelling the chemical changes in wood upon photodegradation?

Answer: Thank you for these valuable and encouraging comments. In that sense, our attempt to draw attention on the used method might also be useful to the researchers from the other fields of the wood science. Practical conclusions have been added and it seems that this technique can be applied to modelling the chemical changes in wood surface upon photodegradation. 

Comment: There is a lot of minor linguistic mistakes (including punctuation, “a/an” and “the” articles missing etc.) that should be corrected, e.g.:

Line 31-32 – „..report on and mechanical properties…” – remove „and”,

Line 33 – “components” instead of “composites”,

Line 40 – “extractives” instead of “extract materials”,

Line 42 – “affect” instead of “effect”,

Line 43-44 – “the color of wood” instead of “the color of wood species” etc.

Answer: These and some other linguistic mistakes has been corrected.

Round 2

Reviewer 1 Report

The authors took into their consideration my comments and the manuscript was greatly improved. However, I still insist that a statistical analysis (analysis of variance or t-test) of the raw data presented, should be performed.

Author Response

Response to reviewer (Ms. Ref. No.: Coatings-735878)

The manuscript was revised according to 1. reviewer’s suggestions. In the text of the revised manuscript the changes made are marked in red.

Reviewer#1:

Comment: The authors took into their consideration my comments and the manuscript was greatly improved. However, I still insist that a statistical analysis (analysis of variance or t-test) of the raw data presented, should be performed.

Answer: The statistical tests has been added to the manuscript.